# Direct Neuronal Reprogramming of Common Marmoset Fibroblasts by ASCL1, microRNA-9/9*, and microRNA-124 Overexpression

**DOI:** 10.3390/cells10010006

**Published:** 2020-12-22

**Authors:** Akisa Nemoto, Reona Kobayashi, Sho Yoshimatsu, Yuta Sato, Takahiro Kondo, Andrew S. Yoo, Seiji Shiozawa, Hideyuki Okano

**Affiliations:** 1Department of Physiology, School of Medicine, Keio University, Shinjuku-ku, Tokyo 160-8582, Japan; aki_nemo@keio.jp (A.N.); reona_kobayashi@keio.jp (R.K.); yoshima@a7.keio.jp (S.Y.); tkondo@keio.jp (T.K.); shiozawa@a7.keio.jp (S.S.); 2Laboratory for Marmoset Neural Architecture, RIKEN Center for Brain Science, Wako City, Saitama 351-0198, Japan; y.sato@brain.bio.keio.ac.jp; 3Laboratory for Proteolytic Neuroscience, RIKEN Center for Brain Science, Wako City, Saitama 351-0198, Japan; 4Graduate School of Science and Technology, Keio University, Kanagawa 223-8522, Japan; 5Department of Developmental Biology, Washington University School of Medicine, St. Louis, MO 63110, USA; yooa@wustl.edu; 6Institute of Animal Experimentation, School of Medicine, Kurume University, Fukuoka 830-0011, Japan

**Keywords:** direct reprogramming, induced neuron (iN), common marmoset, microRNA-9/9*, microRNA-124, ASCL1

## Abstract

The common marmoset (*Callithrix jacchus*) has attracted considerable attention, especially in the biomedical science and neuroscience research fields, because of its potential to recapitulate the complex and multidimensional phenotypes of human diseases, and several neurodegenerative transgenic models have been reported. However, there remain several issues as (i) it takes years to generate late-onset disease models, and (ii) the onset age and severity of phenotypes can vary among individuals due to differences in genetic background. In the present study, we established an efficient and rapid direct neuronal induction method (induced neurons; iNs) from embryonic and adult marmoset fibroblasts to investigate cellular-level phenotypes in the marmoset brain in vitro. We overexpressed reprogramming effectors, i.e., microRNA-9/9*, microRNA-124, and Achaete-Scute family bHLH transcription factor 1, in fibroblasts with a small molecule cocktail that facilitates neuronal induction. The resultant iNs from embryonic and adult marmoset fibroblasts showed neuronal characteristics within two weeks, including neuron-specific gene expression and spontaneous neuronal activity. As directly reprogrammed neurons have been shown to model neurodegenerative disorders, the neuronal reprogramming of marmoset fibroblasts may offer new tools for investigating neurological phenotypes associated with disease progression in non-human primate neurological disease models.

## 1. Introduction

The common marmoset (*Callithrix jacchus*), a small and fecund non-human primate, has attracted attention in the field of neuroscience research due to its anatomical and behavioral similarities to humans. Especially, the availability of transgenic techniques has allowed genetically modified marmosets to be developed as human disease models [1,2,3,4]. Although these studies have proven the significance of marmoset models, the following challenges remain: (i) marmoset models for late-onset diseases require several years to develop disease phenotypes because of their long lifespan (16–21 years [5]), and (ii) the onset age and severity of phenotypes can vary among individuals due to differences in genetic background, making it difficult to predict them before onset. Direct brain tissue sampling from live marmosets is highly invasive and interferes with behavioral analysis. To overcome these problems, a low-invasive and non-intrusive evaluation method is required.

Somatic cells in a terminally differentiated state can be reprogrammed into pluripotent stem cells by the overexpression of defined transcription factors, suggesting that artificial reversion of cell fate is feasible [6]. Previous studies have reported that several transcription factors can directly convert mouse and human somatic cells into various types of cells, such as cardiomyocytes [7,8], hepatocytes [9,10], neural stem cells [11], and neurons [12,13]. Among the reprogramming techniques utilizing induced pluripotent stem cells, the direct conversion of somatic cells into neurons (induced neurons; iNs) has advantages for in vitro disease modeling of neurodegenerative diseases in terms of its rapid and tractable induction. The iN method also permits neuron-induction while retaining age-related epigenetic signatures [14]. Thus, we focused on the direct neuronal reprogramming of marmoset somatic cells.

We previously reported a direct neuronal induction method from embryonic marmoset fibroblasts by overexpressing four transcription factors: POU class 3 homeobox 2, Achaete-Scute family bHLH transcription factor 1 (ASCL1), myelin transcription factor 1-like, and neuronal differentiation 1 [15]. However, there are serious problems associated with this method, such as the low-induction efficiency (~1%), insufficient maturation, and inability to induce neurons from postnatal marmoset fibroblasts, which may impose limitations on its applicability for the in vitro analysis of genetically modified marmosets.

Recent studies on human fibroblasts have revealed a new approach for the direct conversion of fibroblasts to neurons using microRNA-9/9* and microRNA-124 (miR-9/9*-124) [16,17], which inhibit the neuronal gene-specific repressor REST complex and promote neuronal differentiation and maturation [18]. Furthermore, ASCL1, a key transcription factor of neuronal specification in early development [19] can also directly induce the conversion of fibroblasts into neurons. In the present study, we demonstrated that the exogenous expression of miR-9/9*-124 and ASCL1 could induce embryonic and adult marmoset fibroblasts to develop into neuronal cells. The iNs generated in this study showed neuronal gene expression and spontaneous neuronal activity, as measured by calcium imaging. In addition, the conversion of marmoset somatic cells into neuron-like cells occurred within only two weeks, in contrast to the extended culture time of months to differentiate induced pluripotent stem cells into neurons. Importantly, the iN conversion method could also be extended to postnatal fibroblasts, which is critical for the in vitro evaluation of genetically modified marmoset phenotypes without surgical invasion in the fetal period.

## 2. Materials and Methods

### 2.1. Animals and Ethical Statements

Common marmoset tissue samples were collected at the RIKEN Center for Brain Science. The animal experiments in this project were approved by the Animal Experimental Committee of RIKEN (approval number H29-2-243(5)), and were performed in accordance with institutional guidelines for the use of laboratory animals at RIKEN, which are consistent with the Guidelines for the Proper Conduct of Animal Experiments by the Science Council of Japan (2006). Animal care was conducted in accordance with the recommendations of the Guide for the Care and Use of Laboratory Animals (National Research Council, 2011). The two adult marmosets used in the current study were 2 years and 4–5 months old when their ear skin-derived fibroblasts were collected. These adult marmosets were born from independent parents. For embryonic fibroblasts, dorsal skin-derived fibroblasts from two littermate marmosets at embryonic day 95 were used. Information on the age, sex, and individual identification number of each marmoset is listed in Appendix A.

### 2.2. Vectors

The piggyBac transposon vector (pCMV-hyPBase) was kindly provided by Drs. Kosuke Yusa and Allan Bradley (Wellcome Trust Sanger Institute). pPB-tetpA (hBCL2-miR-9/9*-124)-iRPT was obtained with the Gateway LR reaction using the Gateway LR Clonase II Enzyme mix (Thermo Fisher Scientific, Waltham, MA, USA), pENTR-hBCL2 and pENTR-R2-mmu-miR-9/9*-124-L4 (both were kindly provided by Dr. Takefumi Sone, Keio University), and pPB-tetpA(R1R4)-iRPT. To generate pPB-tetpA(R1R4)-iRPT, the *att*R2 site of pPB-tetpA-iRPT [20] was replaced with an *att*R4 site by seamless cloning using a GeneArt Seamless PLUS Cloning and Assembly Kit (Thermo Fisher Scientific) and an *att*R4 fragment amplified from pUC-DEST-R3R4(R) [21] following *Eco*RI and *Sal*I digestion. The CSIV-TRE-ASCL1 lentiviral vector was constructed by the Gateway LR reaction of CSIV-TRE-DEST (kindly provided by Dr. Takefumi Sone) and pENTR-hASCL1 (kindly provided by Drs. Yuhki Nakatake and Minoru Ko, Keio University). We have deposited pPB-tetpA(R1R4)-iRPT (#154882), pPB-tetpA(hBCL2-miR-9/9*-124)-iRPT (#154881), and CSIV-TRE-ASCL1 (#154883) into Addgene (https://www.addgene.org). For lentivirus production, pCAG-HIVgp and pCMV-VSV-G-RSV-Rev were kindly provided by Dr. Hiroyuki Miyoshi (Keio University).

### 2.3. Lentivirus Production

The TRE-ASCL1 lentivirus was produced as described previously [15]. In brief, 3.0 μg pCAG-HIVgp, 3.0 μg pCMV-VSV-G-RSV-Rev, and 6.0 μg CSIV-TRE-ASCL1 were transfected into HEK293T cells by polyethyleneimine in a 10-cm sub-confluent dish in Dulbecco’s modified Eagle’s medium with high glucose (DMEM; Thermo Fisher Scientific) supplemented with 10% inactivated fetal bovine serum (FBS; Sigma Aldrich, St. Louis, MO, USA) and 1% penicillin/streptomycin solution (Nacalai Tesque, Kyoto, Japan). The following day, the medium was replaced with 10% FBS DMEM containing 10 µM forskolin (Sigma Aldrich). At 2 days after medium change, the lentivirus-containing supernatant was collected and passed through a 0.2-µm filter. The filtrated supernatant was ultracentrifuged at 25,000 rpm for 2 h at 4 °C. The ultracentrifuged supernatant was additionally concentrated by ultrafiltration using Amicon Ultra 0.5-mL filters (Millipore, Burlington, MA, USA).

### 2.4. Cell Culture and Transfection

Fibroblasts were cultured in fibroblast medium consisting of DMEM supplemented with 10% inactivated FBS and 1% penicillin/streptomycin solution. The day before transfection, the cells were seeded on a gelatin-coated 6-well plate (Greiner, Monroe, NC, USA) and incubated overnight at 37 °C in 5% CO_2_. For transfection, 1.0 μg DNA vectors, consisting of 0.8 μg pPB-tetpA(hBCL2-miR-9/9*-124)-iRPT and 0.2 μg pCMV-hyPBase, lipofectamine-LTX PLUS reagent (1 μL; Thermo Fisher Scientific), and LTX reagent (2.5 μL; Thermo Fisher Scientific) were diluted in 200 μL Opti-MEM (Thermo Fisher Scientific) and added to the sub-confluent cells in each well of a 6-well plate. The following day, fresh fibroblast medium was added. At 3 days after transfection, puromycin selection (5 μg/mL) was started and conducted for 1–2 weeks. After selection, the cells were infected with the TRE-ASCL1 lentivirus.

### 2.5. Neuronal Induction

Fibroblasts transfected with pPB-tetpA(hBCL2-miR-9/9*-124)-iRPT and the TRE-ASCL1 lentivirus were used for neuronal conversion. On the first day of induction (post induction day [PID] 0), 1.0 × 10^5^ cells were seeded in each well of a gelatin-coated 12-well plate (Iwaki, Shizuoka, Japan) and cultured in fibroblast medium containing 2 μg/mL doxycycline (Dox; FUJIFILM Wako Pure Chemical, Osaka, Japan). On PID 2, the cells were cultured in fresh fibroblast medium containing Dox. On PID 4, a 50-μL cell drop consisting of fibroblast medium and 5.0 × 10^4^ cells was added to each well of a poly-L-ornithine/laminin/fibronectin-coated plate, and cultured in fibroblast medium after the cells attached. The following day (PID 5), the medium was switched to a neuronal reprogramming medium composed of a 1:1 mixture of Neurobasal medium (Thermo Fisher Scientific) and DMEM/F-12 (FUJIFIM Wako Pure Chemical) supplemented with 1% N2 supplement (Thermo Fisher Scientific), 2% B27 supplement (Thermo Fisher Scientific), 1% non-essential amino acid solution (Sigma Aldrich), 1 mM L-glutamine (Nacalai Tesque), 200 μM dbcAMP (Sigma Aldrich), 10 μM forskolin, 10 μM Y-27632 (FUJIFILM Wako Pure Chemical), 3 μM CHIR99021 (Axon Medchem, Groningen, Nederland), 1 μM PD-0325901 (FUJIFILM Wako Pure Chemical), 0.5 μM A83-01 (Santa Cruz, Dallas, TX, USA), 10 ng/mL recombinant human leukemia inhibitory factor (Oriental Yeast, Tokyo, Japan), 1 mM valproic acid (FUJIFILM Wako Pure Chemical), 10 ng/mL brain-derived neurotrophic factor (Peprotech, Rocky Hill, NJ, USA), 10 ng/mL glial cell line-derived neurotrophic factor (Peprotech), 10 ng/mL neurotrophin-3 (Peprotech), 200 μM ascorbic acid (Sigma Aldrich), and 1 μM retinoic acid (Sigma Aldrich), supplemented with Dox and penicillin/streptomycin. The medium was changed every 4 days.

### 2.6. Immunocytochemistry

The cells were fixed in 4% paraformaldehyde for 15 min and permeabilized with 0.3% Triton-X in phosphate-buffered saline (PBS) for 10 min. The cells were blocked with 5% bovine serum albumin (Nacalai Tesque) in PBS for 1 h and incubated overnight at 4 °C with the following primary antibodies: rabbit anti-class III β-tubulin (βIII-tubulin; Abcam; ab18207; 1:500), mouse anti-βIII-tubulin (Sigma Aldrich; T8660; 1:500), mouse anti-microtubule associated protein 2 (MAP2; Sigma Aldrich; M4403; 1:500), mouse anti-NeuN (Merck Millipore; MAB377; 1:500), guinea pig anti-synaptophysin (SYP) 1 (Synaptic Systems; 101 004; 1:500), rabbit anti-postsynaptic density protein 95 (PSD95; Thermo Fisher Scientific; 51-6900; 1:500), rabbit anti-vesicular glutamate transporter 1 (Synaptic Systems; 135 303; 1:500), and mouse anti-ASCL1 (BD Pharmingen; 556604; 1:500). The following day, the cells were washed with PBS and incubated with secondary antibodies conjugated with Alexa Fluor 488, 555, or 647 (Thermo Fisher Scientific; 1:1000) for 1 h at room temperature. The nuclei were stained with Hoechst 33342 (Thermo Fisher Scientific; 1:2000). When immunostaining with Alexa Fluor 647, the cells were heat-treated at 105 °C for 5 min in Target Retrieval Solution (Agilent, Santa Clara, CA, USA) after fixation to extinguish iRFP fluorescence. Images were observed under the LSM 700 confocal system (Carl Zeiss, Oberkochen, Germany).

### 2.7. Calcium Imaging Analysis

The cells were washed twice with PBS and incubated in the neuronal reprogramming medium containing Fluo-8 (AAT Bioquest, Sunnyvale, CA, USA; 1 μM) and Hoechst 33342 (1:1000) for 20 min at 37 °C in 5% CO_2_. After incubation, the cells were washed twice with PBS and fed with neuronal reprogramming medium. For tetrodotoxin treatment, the reagent was dropped into the culture wells to reach a final concentration of 5 µM. Movies were captured at 20 frames/s using an IX83 inverted microscope (Olympus, Tokyo, Japan) equipped with an electron multiplying CCD camera (Hamamatsu Photonics, Hamamatsu, Shizuoka, Japan) and pE-4000 LED illumination system (CoolLED, Andover, UK). Imaging data analysis was performed using the miniscope 1-photon-based calcium imaging signal extraction pipeline [22], which contains modules for background subtraction, movement correction, and neural signal extraction. After that, neuron locations and traces were extracted, and we identified the rising phase of each calcium transient as the calcium event (peak ΔF/F_0_ > 0.5 standard deviations of baseline fluctuations). The start of this rising phase is detected when the 1st derivative of ΔF/F_0_ rises above 0 and continues to increase above 2 standard deviations of baseline fluctuations.

### 2.8. Quantitative RT-PCR

RNA was isolated using an RNeasy Mini Kit (QIAGEN, Hilden, Germany) according to the manufacturer’s protocol. Total RNA (1.0 μg) was reverse-transcribed in ReverTra Ace qPCR RT master mix (Toyobo, Osaka, Japan). The resultant cDNAs were diluted to 4 ng/μL in nuclease-free water. Quantitative RT-PCR was performed using TB Green Premix Ex Taq II (TaKaRa Bio, Kusatsu, Shiga, Japan) on the ViiA 7 Real-Time PCR System (Thermo Fisher Scientific) according to the manufacturer’s instructions. The sequences of primers used in this study are listed in Appendix A. Glyceraldehyde 3-phosphate dehydrogenase expression was used for normalization. Four-year-old marmoset frontal cortex (brain tissue) was used for control samples.

### 2.9. Tissue Preparation

The marmoset brain tissue used for control sample was obtained from a four-year-old male marmoset. To obtain brain tissue, the marmoset was anesthetized with ketamine (10 mg/kg) and pentobarbital (80 mg/kg) by intramuscular injection. Under deep anesthesia, the marmosets were euthanized by exsanguination and perfused intracardially with 0.1 M phosphate-buffered saline (PBS, pH 7.6). After 10 min of perfusion, the frontal cortex was removed, immediately frozen in liquid nitrogen, and stored at −80 °C until RNA extraction as described above.

### 2.10. Transcriptomic Analysis

Extraction of total cellular RNA was performed as described above. Poly(A)+ RNA was selected and converted to a library of cDNA fragments (mean length: 350 bp) with adaptors attached to both ends for sequencing using a KAPA mRNA Capture Kit (KK8440; Kapa Biosystems, Wilmington, MA, USA), KAPA RNA HyperPrep Kit (KK8542; Kapa Biosystems), KAPA Pure Beads (KK8543; Kapa Biosystems), and SeqCap Adapter Kit A (Roche, Basel, Switzerland) according to the manufacturers’ instructions. The cDNA libraries were quantified using a KAPA Library Quantification Kits (KK4828; Kapa Biosystems), followed by sequencing using an Illumina HiSeq X to obtain 150-nucleotide sequences (paired-end). RNA-sequencing (seq) data (*fastq* file format) were quality checked, and low-quality reads (score < 30), adapter sequences, and overrepresented sequences such as poly-A chains were trimmed using *Trim Galore!* (ver.0.4.0). The remaining reads were mapped to the *C. jacchus* genome (cj3.2.1.86) using *STAR* (ver.2.5.3a) [23], and the output file (BAM file format) was summarized using *featureCounts* [24] (1.5.2). The summarized data were processed using *DESeq2* [25] to estimate their size factors, followed by the removal of reads not expressed in any of the samples. We identified differentially expressed genes with a cutoff of 0.01 for Benjamini-Hochberg-adjusted *p*-values and a cutoff of 5 for the fold-change ratio. Principal component analysis was performed using variance-stabilizing transformation of the estimated counts. Enrichment analysis using gene ontology was performed using *clusterProfiler* [26]. Heatmaps of gene expression were drawn using the row-wise z-scores of the variance-stabilizing transformation-normalized expression values of each gene. We have deposited the raw and processed RNA-seq data obtained in this study into the Gene Expression Omnibus (accession number: GSE152433). For the analysis, we also included deposited data (GSE152264) obtained in our previous study [20].

### 2.11. Statistical Analysis

All data are expressed as the mean ± standard error of the mean. Statistical analyses were performed using RStudio. mRNA expression levels in each group were compared statistically with one-way analysis of variance, followed by a post hoc Tukey’s test.

## 3. Results

### 3.1. Direct Neuronal Induction of Adult and Embryonic Somatic Fibroblasts Derived from Marmosets

We previously reported a direct neuronal induction method using marmoset embryonic skin fibroblasts by lentiviral overexpression of four transcription factors [15], but with low induction efficiency (~1%) and inapplicability to adult marmoset fibroblasts. Accordingly, the aim of the present study was to devise a novel neuronal induction method that is efficient and applicable to adult marmoset fibroblasts. For that purpose, we focused on neuron-specific microRNAs (miR-9/9*-124) that have been demonstrated to induce the efficient neuronal conversion of human fibroblasts [16,17]. Moreover, we tested the combined overexpression of the neuronal transcription factor *ASCL1*, a pioneer transcription factor for the direct neuronal conversion of mouse and human iNs [27,28], with these microRNAs for efficient induction.

We first constructed a piggyBac transposition-based vector [29] encoding miR-9/9*-124 and the anti-apoptotic factor human B-cell lymphoma 2 (BCL2) under Dox-dependent transcriptional control by the tetracycline response element (TRE). In addition to its role in apoptosis inhibition, BCL2 promotes the neuronal conversion of somatic cells by enhancing mitochondrial oxidative metabolism [30]. The piggyBac vector also carried infrared fluorescent protein (iRFP), a drug selection marker (puromycin N-acetyl-transferase gene), and reverse tetracycline transactivator under the control of the elongation factor 1α promoter in the opposite direction to miR-9/9*-124 (Figure 1A, left). Each gene was connected by 2A peptides. We also constructed a lentiviral vector carrying *ASCL1* under the control of the TRE (Figure 1A, right) for its combined overexpression with miR-9/9*-124 in a Dox-dependent manner. Both vectors were co-transfected into marmoset fibroblasts, followed by drug selection using puromycin for 1–2 weeks. Successful transduction was confirmed by expression of the iRFP reporter in almost all cells (Appendix A). After drug selection, each fibroblast cell line derived from two embryonic and adult marmosets (Appendix A) was induced to express miR-9/9*-124 and ASCL1 by Dox treatment from PID 0, as shown in the timetable in Figure 1B. These fibroblast cells were re-plated on a dish with neuronal cell-optimized coating at PID 4. The fibroblast medium was changed to neuronal reprogramming medium at PID 5, and the fibroblasts then underwent a morphological change into a spherical structure within a few hours. Moreover, the spherical cells extended neurite-like projections and changed to a neuron-like morphology at two weeks after the start of induction (Figure 1C). There were no morphological differences between the neuron-like cells derived from embryonic and adult marmoset fibroblasts. These results showed that miR-9/9*-124 and *ASCL1* overexpression can convert adult and embryonic marmoset fibroblasts into neuron-like cells rapidly.

### 3.2. Neuron-Like Cells Express a Subset of Pan-Neuronal Markers at the Protein Level

To characterize the neuron-like cells, we performed immunocytochemistry using antibodies for pan-neuronal markers. We found that the neuron-like cells derived from embryonic and adult marmoset fibroblasts expressed pan-neuronal cytoskeletal markers, such as neuron-specific βIII-tubulin and MAP2, in the neurite-like projections and cell bodies at PID 15 (Figure 2A). Furthermore, the expression of the mature neuronal nuclear marker NeuN was also detected in the neuron-like cells (Figure 2A). These neuronal markers were not expressed in the original fibroblasts (Appendix A). Quantitatively, ~80% of the neuron-like cells were positive for βIII-tubulin and a similar proportion of cells were positive for MAP2 (Figure 2B). In addition, ~60% of the neuron-like cells derived from adult marmoset fibroblasts and ~80% of the cells derived from embryonic fibroblasts were positive for NeuN (Figure 2B). The expression of the synaptic markers SYP and PSD95 was also detected in the embryonic and adult neuron-like cells (Figure 2C). Inspection of high-magnification images revealed the appearance of immuno-positive puncta along the periphery of the neurite-like structures (Figure 2D). The colocalization of pre- and postsynaptic puncta was sometimes observed in the neuron-like cells derived from embryonic marmoset fibroblasts. A comparison of the number of SYP- and PSD95-positive puncta per mm^2^ showed a higher density of synaptic puncta in the embryonic neuron-like cells than in the adult neuron-like cells (Figure 2E). Some of the neuron-like cells were positive for the excitatory synaptic marker vesicular glutamate transporter 1, especially in the embryonic fibroblast-derived neuron-like cells with a high frequency (Figure 2F). We found that the ASCL1-expressing neuron-like cells were positive for βIII-tubulin (Appendix A, arrow), whereas the ASCL1-negative cells did not express it (arrowhead), suggesting a contribution of ASCL1 to neuronal induction. These results demonstrated that the combined overexpression of miR-9/9*-124 with *ASCL1* can induce the expression of neuronal proteins.

### 3.3. Neuron-Like Cells from Embryonic and Adult Fibroblasts Show Neural Activity

To evaluate the neuronal functionality of the neuron-like cells, we performed calcium imaging analysis using the calcium indicator dye Fluo-8 to monitor intracellular calcium dynamics. Calcium transients were not observed in the induced cells soon after induction (PID 1); however, after the morphological changes occurred (at approximately PID 10), we observed spontaneous activity in the neuron-like cells derived from embryonic and adult fibroblasts (Figure 3A, Appendix A). Interestingly, we observed synchronized activity among large numbers of neuron-like cells derived from embryonic fibroblasts, but not in those derived from adult fibroblasts (Figure 3B). This synchronized activity may have occurred via synaptic interactions, considering the expression of synaptic markers (Figure 2C–E). In addition, we assessed the neuronal function via drug-induced modulation, and found that the calcium transients were inhibited by the sodium channel blocker tetrodotoxin (Figure 3C). Overall, these data showing that the neuron-like cells from embryonic and adult fibroblasts have neuronal function indicate these cells successfully differentiated into functional neurons.

### 3.4. Transcriptomic Analysis of Neuron-Like Cells

To analyze the global gene expression profiles of the neuron-like cells, we performed RNA-seq and compared the profiles between the original fibroblasts and neuron-like cells. Principal component analysis indicated segregation between the original fibroblasts and neuron-like cells (Figure 4A). Our gene expression analysis also showed the upregulation of neuronal marker genes and the downregulation of fibroblast marker genes (Figure 4B). For instance, the gene expression of pan-neuronal markers such as *MAP2*, RNA binding fox-1 homolog 3 (*RBFOX3*), microtubule-associated protein tau (*MAPT*), and synaptotagmin 1 (*SYT1*) was upregulated, while the gene expression of fibroblast markers such as collagen type I alpha 1 chain (*COL1A1*), collagen type III alpha 1 chain (*COL3A1*), and vimentin (*VIM*) was downregulated in the neuron-like cells. Analysis of the top five gene ontology terms revealed that the upregulated genes in the neuron-like cells were enriched with terms related to neuronal differentiation and development, while the downregulated genes were associated with fibroblast function (Figure 4C). Consistent with the results of the protein expression levels of SYP and PSD95 between the embryonic and adult neuron-like cells (Figure 2C–E), the mRNA expression of several synaptic markers was higher in the embryonic neuron-like cells than in the adult neuron-like cells (Appendix A). Moreover, neuronal subtype-specific gene markers were not upregulated in the neuron-like cells (Appendix A).

We also found that some of the targets of miR-9/9*-124 were regulated in our induction method, e.g., polypyrimidine tract-binding protein 1/2 (*PTBP1/2*). In somatic fibroblasts, *PTBP1* represses neuronal conversion [31], while *PTBP2* accelerates neuronal conversion [32]. Our gene expression analysis showed the upregulation of *PTBP2* and downregulation of *PTBP1* in the neuron-like cells (Figure 4B).

To evaluate further the neuronal transcripts of the neuron-like cells, we performed quantitative RT-PCR analysis using cDNAs from the original fibroblasts, neuron-like cells, and marmoset brain tissue. Initially, we quantified the gene expression levels of pan-neuronal markers, such as *TUBB3* (encoding βIII-tubulin), *MAP2*, *RBFOX3*, and *MAPT*. The results showed the elevation of their gene expression in the neuron-like cells compared to the original fibroblasts, and their expression levels were comparable with those of marmoset brain tissue (Figure 4D). In contrast, the expression of the fibroblast marker gene *COL1A1* was significantly decreased in the neuron-like cells compared to the original fibroblasts (Figure 4E), and was barely detectable in marmoset brain tissue. These results indicate that the neuronal induction of fibroblasts by miR-9/9*-124 and ASCL1 leads to an increase in the expression of several neuronal marker genes and a decrease in the expression of fibroblast marker genes.

## 4. Discussion

Although we previously reported a direct neuronal induction method for embryonic marmoset fibroblasts, the induction of neuronal cells using fibroblasts derived from postnatal marmosets was not achieved [15]. Here, we established a rapid, efficient, and robust method for direct neuronal conversion, which is applicable to somatic fibroblasts derived from embryonic and adult marmosets. We used a previously reported method based on microRNAs (miR-9/9*-124) and a transcription factor (ASCL1), which directly convert human fibroblasts into neurons. We demonstrated that the combination of these factors can also induce neuron-like cells from marmoset fibroblasts with high efficiency (about 80%, Figure 2B). The resultant neuron-like cells showed typical features of neurons, such as neuron-like morphology and the expression of neuronal proteins and mRNAs (Figure 1, Figure 2, and Figure 4). Moreover, calcium imaging revealed that the neuron-like cells had the functional properties of neurons (Figure 3). Therefore, these results indicate that the generated neuron-like cells can be defined as marmoset iNs according to the previously reported criteria for defining iNs [33].

Although this newly established marmoset iN method showed a high induction efficiency in adult and embryonic marmoset fibroblasts, some results revealed differences between adult and embryonic iNs. For example, synaptic markers, such as SYP and PSD95, were frequently expressed in the embryonic iNs compared to the adult iNs (Figure 2C–E, Appendix A). These results suggest that embryonic iNs more closely resemble mature neurons than adult iNs, possibly related to previous reports that young-derived cells can be easily reprogrammed at a high efficiency [34]. Considering the negative effect of aging, collecting fibroblasts at a young age is best suited to obtain iNs that are more similar to neurons in brain. However, the fact that the present method enables iN induction from not only embryonic fibroblast but also adult fibroblast extends its usefulness.

Interestingly, we observed synchronized neuronal activity in the embryonic iNs, but not in the adult iNs (Figure 3B). Synaptic interactions are the common mechanism of neural communication during neuronal synchronization. In our study, the embryonic iNs expressed several synaptic markers at a higher level than the adult iNs (Appendix A). Moreover, pre- and postsynaptic marker-positive puncta were partly colocalized in the embryonic iNs (Figure 2D), possibly indicating the formation of synapses. These results suggest the contribution of the synaptic interactions to the synchronized activity observed in the embryonic iNs.

In conclusion, this iN method is broadly applicable for phenotype analysis of disease model marmosets and phenotypic screening to predict the disease severity of pre-symptomatic animals. The embryonic iN is useful for evaluating synaptic dysfunction related to neurodegenerative diseases. Meanwhile, the adult iN method provides neuron-like cells minimally invasively without skin biopsy at the fetal stage, albeit with limitations in the synapse formation. The obtained iNs also allow for the evaluation of neurological dysfunctions related to Parkinson’s disease, as has been reported previously [35,36]. The iN method established in the current study should facilitate progress on neurological disease modeling in non-human primate species.

## Figures and Tables

**Figure 1 cells-10-00006-f001:**
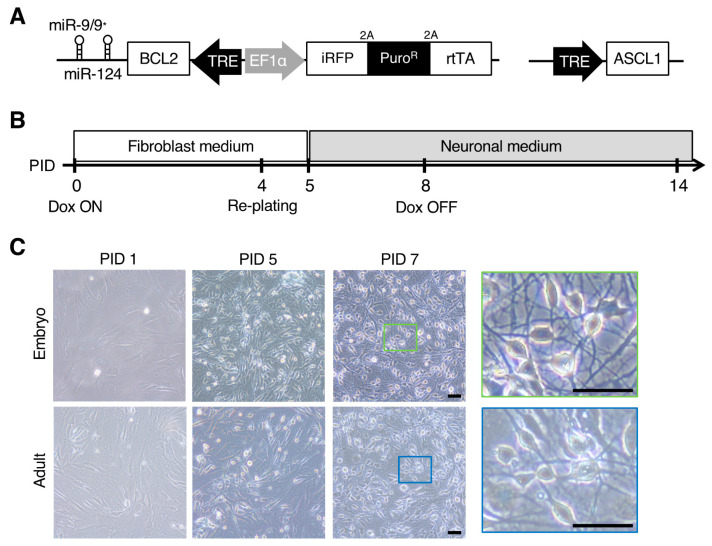
Neuronal induction of marmoset fibroblasts by overexpressing microRNA-9/9* and microRNA-124 (miR-9/9*-124) and Achaete-Scute family bHLH transcription factor 1 (ASCL1). (**A**) Left: PiggyBac transposon vector containing (1) infrared fluorescent protein (iRFP), puromycin resistance (Puro^R^), and reverse tetracycline transactivator (rtTA) under the control of the elongation factor 1α (EF1α) promoter, and (2) human B-cell lymphoma 2 (BCL2) and miR-9/9*-124 under the control of the tetracycline response element (TRE) promoter. Right: Lentiviral vector containing ASCL1 under the control of the TRE promoter. (**B**) Time course for induction. After the transduction of miR-9/9*-124 and *ASCL1*, doxycycline (Dox) was added to start induction. At post induction day (PID) 4, the cells were re-plated on dishes with neuronal cell-optimized coating. The culture medium was replaced with neuronal medium supplemented with small molecules that enhance reprogramming at PID 5. The cells were analyzed within ~2 weeks. (**C**) Morphological changes of neuron-like cells from embryonic and adult fibroblasts during the induction process. The cells retained fibroblast morphology until re-plating at PID 4. After re-plating and neuronal medium change, these cells acquired neuron-like morphology. Spherical cell bodies and neurite-like structures were observed in these cells. Scale bars, 50 µm.

**Figure 2 cells-10-00006-f002:**
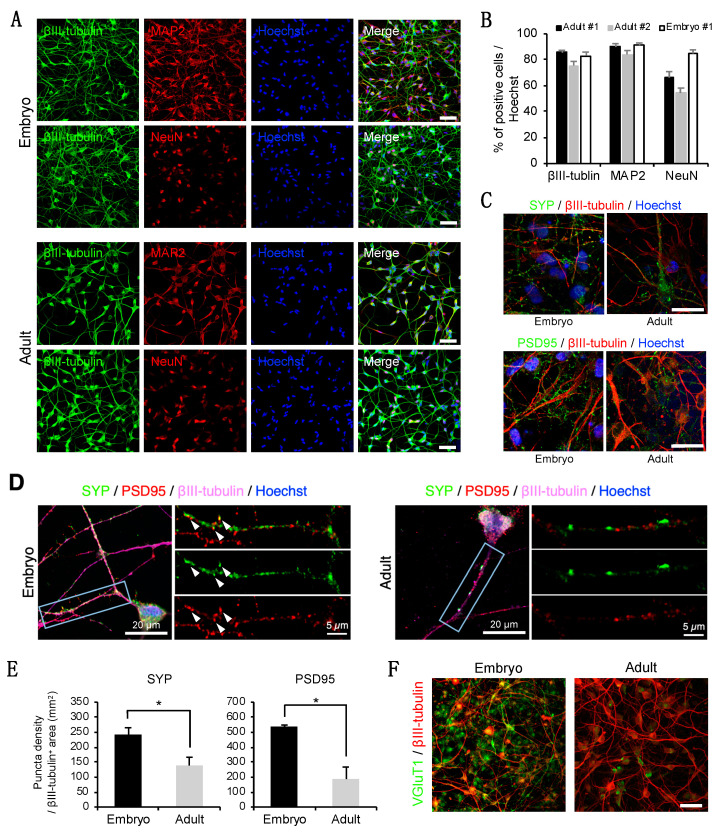
Neuronal marker expression of neuron-like cells by immunocytochemistry. (**A**) Immunostaining for pan-neuronal markers in embryonic and adult marmoset neuron-like cells at post induction day (PID) 15. The cell bodies and neurites were positive for class III β-tubulin (βIII-tubulin) and microtubule-associated protein 2 (MAP2). The nuclei were positive for the mature neuronal marker NeuN. Scale bars, 50 µm. (**B**) Quantification of the number of neuronal marker-positive cells in neuron-like cells derived from fibroblasts from two adult marmosets and an embryonic marmoset (*n* = 3 biological replicates per sample). Data are represented as the mean ± standard error of the mean. (**C**) Co-immunostaining for βIII-tubulin and synaptic markers (synaptophysin [SYP] or postsynaptic density protein 95 [PSD95]). Scale bars, 20 µm. (**D**) Co-immunostaining for βIII-tubulin, SYP, and PSD95. Enlarged images of the neurite area enclosed by the rectangle are shown on the right. The arrowheads indicate the colocalization of the immunostaining-positive puncta. (**E**) Quantification of SYP- and PSD95-immunostained puncta. The number of SYP- or PSD95-positive puncta per βIII-tubulin-positive area (mm^2^) was counted (*n* = 3 biological replicates per sample). Differences between means were compared using Student’s t-test. * *p* < 0.05. (**F**) Immunostaining for the excitatory synaptic marker vesicular glutamate transporter 1 (VGluT1) in embryonic and adult marmoset neuron-like cells at PID 14. Scale bar, 50 µm.

**Figure 3 cells-10-00006-f003:**
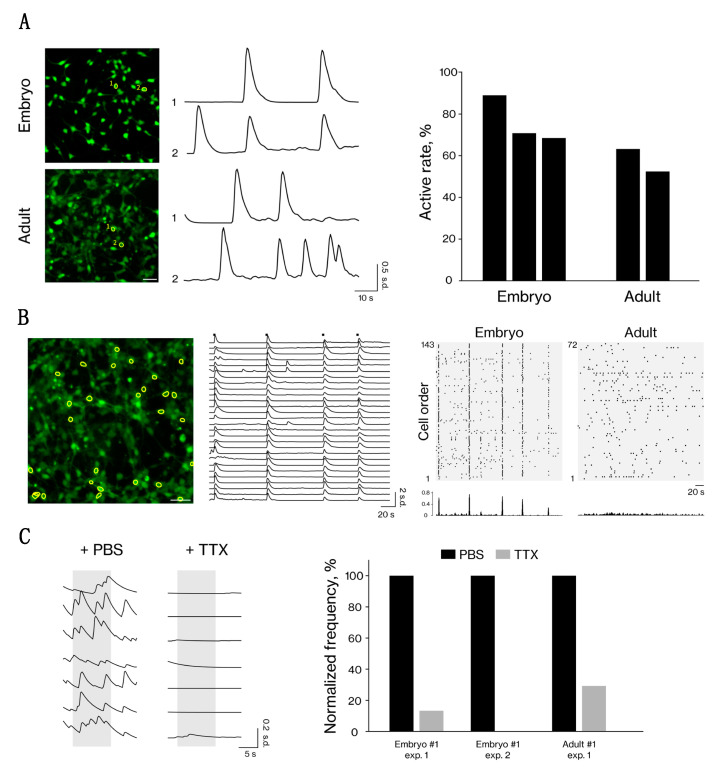
Functional properties of neuron-like cells from embryonic and adult fibroblasts. (**A**) Representative calcium transients of neuron-like cells from embryonic and adult fibroblasts. Left: Images of neuron-like cells in a single field of view, and time traces of the selected cells (highlighted with yellow outlines). Spontaneous neuronal activity was observed in both types of neuron-like cells after induction. Scale bar, 50 µm. Right: The percentage of active cells. The number of Hoechst-positive cells and active cells was calculated, and both types of neuron-like cells showed high activity rates (embryonic: 79.9 ± 9.0%; adult: 57.8 ± 5.4%). Each bar shows independent experiments (*n* = 3 for embryonic neuron-like cells, *n* = 2 for adult neuron-like cells). (**B**) Synchronized activity was detected in the embryonic neuron-like cells. Scale bar, 50 µm. Left: Images of embryonic neuron-like cells in a single field of view. Middle: Time traces for the cells highlighted with yellow outlines. Synchronized activity patterns were observed in the embryonic neuron-like cells. Right: Upper panels show the activity patterns of embryonic and adult neuron-like cells visualized by a raster plot. Black dots indicate calcium events. Notably, synchronized activity patterns were obvious in the embryonic neuron-like cells, but not in the adult neuron-like cells. Lower panels show the histogram of the normalized active cell numbers at each time point. (**C**) Effects of phosphate-buffered saline (PBS) and the sodium channel blocker tetrodotoxin (TTX) on embryonic and adult neuron-like cells. Left: Representative calcium transients of neuron-like cells from embryonic fibroblasts. Gray shading indicates 10–30 s after PBS or TTX administration. Right: Normalized frequencies before and after administrations in embryonic and adult neuron-like cells. Each set of bars shows independent experiments (*n* = 2 for embryonic neuron-like cells, *n* = 1 for adult neuron-like cells).

**Figure 4 cells-10-00006-f004:**
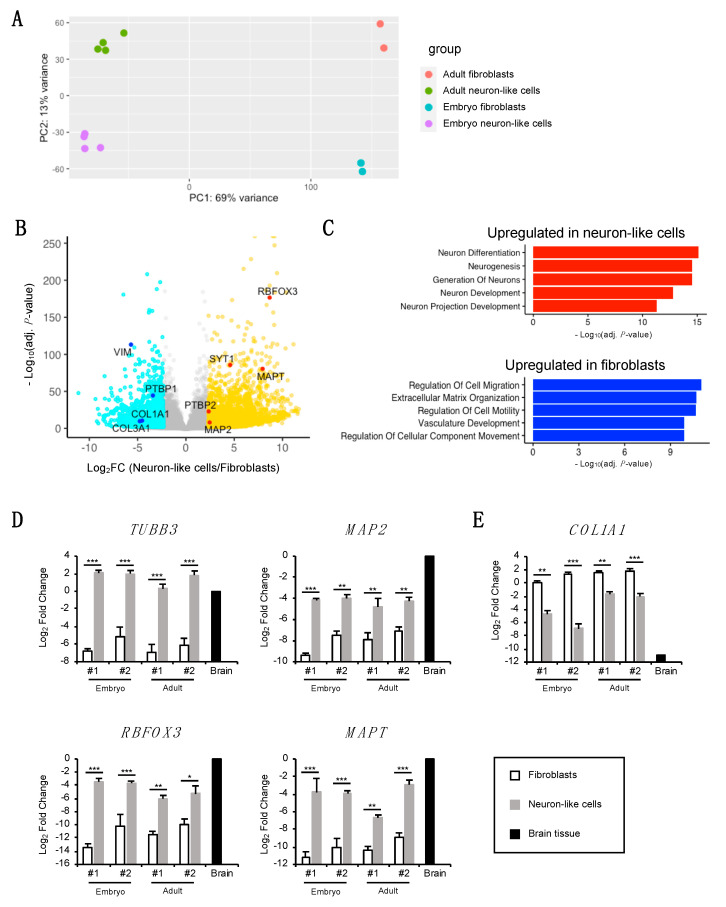
Transcriptomic analysis of neuron-like cells. (**A**) Principal component analysis of gene expression data between the original marmoset fibroblasts (*n* = 1 experiment per animal) and neuron-like cells (*n* = 2 independent experiments per animal). (**B**) Volcano plot representing neuronal and fibroblast marker genes differentially expressed between the original fibroblasts and the neuron-like cells. We identified differentially expressed genes, which are highlighted, with a cutoff of 0.01 for Benjamini-Hochberg adjusted *p*-values and a cutoff of 5 for the fold-change ratio. (**C**) Gene ontology terms associated with the genes upregulated in the neuron-like cells (red) and original fibroblasts (blue). (**D**,**E**) Quantitative PCR analysis of pan-neuronal markers and fibroblast marker genes (normalized to glyceraldehyde 3-phosphate dehydrogenase) in the original fibroblasts (*n* ≥ 3 independent experiments per animal), neuron-like cells (*n* ≥ 3 independent experiments per animal), and brain tissue (from one animal). The *Y*-axis represents the normalized log_2_ fold-change values. Data are represented as the mean ± standard error of the mean. Differences between means were compared using one-way analysis of variance followed by a *post hoc* Tukey’s test. * *p* < 0.05; ** *p* < 0.01; *** *p* < 0.001. COL1A1, collagen type I alpha 1 chain; COL3A1, collagen type III alpha 1 chain; MAP2, microtubule-associated protein 2; MAPT, microtubule-associated protein tau; RBFOX3, RNA binding fox-1 homolog 3; PTBP1/2, polypyrimidine tract-binding protein 1/2; SYT1, synaptotagmin 1; TUBB3, class III beta-tubulin; VIM, vimentin.

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
