# Peer review of "Direct Neuronal Reprogramming of Common Marmoset Fibroblasts by ASCL1, microRNA-9/9*, and microRNA-124 Overexpression"

_cells, 2020, doi:10.3390/cells10010006_

Round 1

Reviewer 1 Report

This paper describes a potentially important new technique for the high efficiency induction of neuron-like cells from both embryonic and adult fibroblasts. This is an improvement on previous techniques that had low efficiency and were only applicable to embryonic fibroblasts. There are some caveats associated with these cells that should be further described in the discussion and additional methodological details regarding the source of cells should be described. Overall, this is a well written paper describing a potentially important technique.

Comments:

  1. Introduction, line 43-44: The maximum lifespan of marmosets in captivity is now reported to be 16-21 years.
  2. Your results seem to indicate that the neuron-like cells that were induced were somewhere along the path between the original fibroblasts and neuron-like cells with the embryonic-origin cells being further along the path towards neurons than the adult-origin cells. Is this accurate? If so, it would be helpful for this to be clearly stated in the discussion.
  3. A clear presentation, perhaps in the form of a table, of similarities and differences between the induced neuron-like cells from embryonic fibroblasts, induced neuron-like cells from the adult fibroblasts, and adult neuronal cells would be an extremely helpful addition.
  4. Please provide additional information regarding the origin of the fibroblasts. For example, were the two adults from the same litter, were they different litter siblings, were they unrelated? Were embryos from a single litter?
  5. Given that the fibroblasts from the two different fibroblast origins led to neuron-like cells that were somewhat different, please include discussion of the advantages and disadvantages of each type and the realistic potential use of each type of cell.
  6. There are likely limitations on the use of these cells for studies of neurodegeneration given the age and therefore aging profile of the cells from which they were derived. Do you think you would be as successful inducing neurons from fibroblasts from older animals (for example those over 8 years of age)? Please include discussion of the limitation for neurodegeneration studies based on the original age of the fibroblasts.
  7. Marmoset brain tissue was used to compare gene expression levels of the neuron-like cells. How was this tissue acquired? Was this from adults, what ages? How many animals/replicates? Please provide information on the origin and acquisition of these tissues.

Reviewer 2 Report

In this report, the Nemoto et al. describe the direct neural conversion of marmoset fetal and adult fibroblasts by overexpression of microRNAs and ASCL1. The authors were able to establish a rapid and robust method for neural induction and improved their results relative to previous studies. The report presents new results on the direct neural conversion of marmoset cells and would be of interest to scientists working in the field. However, there are some points that need to be clarified before it could be considered for publication:

- The results of the previous work [15] yielded low efficiency of neural induction (~1%) and inapplicability to adult marmoset fibroblasts. However, it is not clear what the efficiency was in the current study. Please, clarify this question so that it would be possible to compare the neural induction efficiency with the older study and between the iN cells generated from fetal and adult fibroblasts.

- It is not clear how many biological replicates were generated from each of the 2 embryonic and 2 adult fibroblast lines. From Figure 4 can be seen that at least 3 biological replicates per group were used for qPCR, but it is not clear how many were obtained from the fetal and the adult fibroblasts. This important information should be included in the results.

- The results from the transcriptome analysis indicate that pan-neuronal markers were up-regulated in the iN cells. How about any more specific markers that would indicate whether the iN cells are oriented more towards any particulat neuronal area (forebrain, midbrain or hindbrain) or particular neuronal subype(s) (serotonergic, dopaminergic, GABAergic, etc)? If any of these markers were upregulated in the iN cells, an overview should be included in the results.

Minor comments:

- Lines 228-231: „Both vectors were co-transfected into marmoset fibroblasts, followed by drug selection using puromycin for 1−2 weeks. Successful transduction was confirmed by expression of the iRFP reporter in almost all cells (Supplementary Figure S1). Therefore, this stable cell line was used for the following neuronal induction experiments. “ - Most readers would find this text confusing, as the wording of the paragraph would suggest that only one line (unknown whether from embryo or adult fibroblasts) was generated for the neural induction. But actually, 2 transgenic  lines from each group were generated and used. Please revise the paragraph.

- Line 361: „(n≥ 3 biological replicates per sample for the original fibroblasts and neuron-like cells).” – under “sample” the authors perhaps mean “experimental group”? Biological replicates belong to an experimental group, while a sample comes from a certain biological replicate.
